# From attributes to value: Neural correlates of a front-of-package label on food decision-making – An fMRI study

Negin Javaheri[1,2]*, Niels Doehring[1,2], Revati Mulay[1,2], Peter Erhard[1],
Manfred Herrmann[1,2]

1 Department of Neuropsychology and Behavioral Neurobiology, University of Bremen, Bremen, Germany,
2 DFG GRK2739/1 Research Training Group: KD²School—Designing Adaptive Systems for Economic Decisions, Germany

* javaheri@uni-bremen.de

## Abstract

Diet-related chronic illnesses, such as obesity and diabetes, pose major global health challenges. To promote healthier choices, policymakers have introduced nudges like front-of-package labels such as the Nutri-Score, which simplify nutritional information. While previous research has examined the impact of front-of-package labels on product valuation via reward and cognitive control pathways, the neural mechanisms underlying attribute-specific changes in food perception (e.g., healthiness, tastiness) remain unclear. In this study, the impact of a Nutri-Score derived color-coded frame on the decision domains , perceived healthiness, perceived tastiness, and willingness-to-pay as a proxy for subjective value, was investigated. Forty healthy participants (28 females, age: M = 23.8 years, SD = 3.1 years) rated 63 food products in two conditions: without (control) and with (treatment) a color-coded frame while undergoing functional magnetic resonance imaging. The investigation focused on how the color-coded frame alters all three decision domains on a behavioral but also on the neural level. Additionally, the variation in neural and behavioral responses depending on the frame color was investigated. Overall, our results show that the color-coded frame significantly influenced WTP, healthiness, and tastiness ratings, with corresponding increases in neural activation in reward-related (ventromedial prefrontal cortex), cognitive control (dorsolateral prefrontal cortex), and homeostatic (thalamus) regions. Healthiness perception involved regions associated with value-based processing (ventromedial prefrontal cortex), as well as cognitive control (dorsolateral prefrontal cortex) when the color-coded frame was present, whereas tastiness perception engaged a network including the insula, brain regions related to reward valuation (ventromedial prefrontal cortex), and regions associated with cognitive control. These findings suggest that the color-coded frame, used here as a proxy for the Nutri-Score, influences food choices by modulating attribute-specific valuation and interacting with homeostatic mechanisms, rather than directly altering overall value computation.

**Data availability statement:** The dataset and associated materials are described in:All relevant data and associated materials are publicly available on the Open Science Framework (OSF): Javaheri, N. (2025, October 30). From attributes to value: The neural impact of a front-of-packaging label on food decision-making – an fMRI study. OSF. https://doi.org/10.17605/OSF.IO/G86ZA.

**Funding:** This work was funded by Deutsche Forschungsgemeinschaft (DFG, German Research Foundation)-GRK2739/1-Project Nr. 447089431-Research Training Group: KD2 School-Designing Adaptive Systems for Economic Decisions.

**Competing interests:** The authors declare that they have no known competing financial or personal interests that could have appeared to influence the work reported in this paper.

Understanding how external cues integrate with internal physiological signals to guide food choices may inform interventions aimed at improving dietary behavior and public health outcomes.

## Introduction

Diet-related diseases, such as obesity and diabetes, are among the leading global health challenges, placing major strain on healthcare systems [1]. In response, governments have implemented nudges, most notably front-of-package labels (FoPLs), to promote healthier food choices [2–5]. FoPLs are typically classified as cognitive nudges [2], while examples such as the Nutri-Score or traffic light labels include also an affective element by using a color-coded grading system. Because of this affective component, the Nutri-Score might also be considered an affective nudge, a category that has been associated with larger effect sizes [2]. Despite extensive research on the effectiveness of FoPLs [5,6], it remains unclear how such FoPL alters the underlying decision-making process during food choices behaviorally and neurally. The present study addresses this gap by leveraging the dual nature of the Nutri-Score and isolating its color-coded component using a color-coded frame derived from each product's Nutri-Score to examine its influence on food-related decision-making.

To investigate the effects on the decision-making process, the study draws on the value-based decision-making theory, which posits that subjective value is computed through the integration of multiple attributes [7]. For instance, when choosing a pack of chips, both its perceived healthiness and tastiness contribute to its overall appeal. Because previous studies have shown that FoPLs can change consumers' food choices [2,8,9] a corresponding shift in willingness-to-pay (WTP), which is a proxy for subjective value, can be expected. Measuring WTP alongside healthiness and tastiness ratings therefore allowed to disentangle how external cues such as color-coded frames shift both attribute perception and overall valuation. Behavioral evidence supports this approach: color-coded labels such as the Nutri-Score reliably increase perceived healthiness of products [10–12] in some cases, affect tastiness ratings, either positively or negatively depending on product category [11,13,14]. Such findings are consistent with the notion that FoPLs influence attribute-level representations, which in turn contribute to the computation of subjective value. Accordingly, if a color-coded frame shifts perceived healthiness or tastiness, it should also shift WTP in the same direction. Grounded in value-based decision-making theory [7] and following our preregistration [15], we formulated the following behavioral hypotheses:

### Hypothesis H1 (Behavioral)

The color-coded frame will systematically modulate subjective valuation (WTP), perceived healthiness, and perceived tastiness. Specifically, compared to control trials, green frames will lead to higher WTP, healthiness ratings, and tastiness ratings, whereas yellow and red frames will lead to lower ratings in all three decision domains.

To further understand the mechanisms through which FoPLs alter food perception and decision-making, we next turn to the neural level of analysis. A recent model of neural activation associated with FoPLs proposes that these labels modulate valuation processes by engaging two principal neural pathways of the core eating network: the ventral reward pathway and the dorsal control pathway [16]. According to this neural FoPL model, nutritional labels indicating healthy food products (e.g., Nutri-Score level "A", or green traffic light labels) predominantly engage the ventral reward pathway, including the ventromedial prefrontal cortex (vmPFC) and ventral striatum, which are critical for value computation and reward processing [17,18]. Moreover, it has also been shown that green traffic light labels indicating healthy food products enhance functional coupling between the vmPFC and posterior cingulate cortex (PCC), reinforcing reward-based valuation [19] in comparison to a guideline daily amount (GDA) label, which lack affective color components. In contrast, labels signaling unhealthy food products through the red color (e.g., Nutri-Score level "E", or red traffic light labels) activate not only the ventral reward pathway, but also the dorsal control pathway. Previous studies comparing red traffic light labels, which include an affective element through color, with GDA labels have reported activation in regions such as the dorsolateral prefrontal cortex (dlPFC) and anterior cingulate cortex (ACC), both associated with self-regulation and inhibitory control [19,20]. Taken together, dietary decision outcomes appear to be shaped by the interplay between reward and control systems. This finding is consistent with the common currency theory, which states that values, irrespective of what kind of decision, are computed in the vmPFC [21].

However, the neural FoPL model focuses on overall value computation and does not explain how individual attributes, such as perceived healthiness and tastiness, contribute to valuation. Although substantial research has investigated the neural correlates of overall valuation [7,21,22], far less is known about how these distinct attributes are processed on their own [23,24], how external cues like FoPLs modulate their perception [25], and the neural pathways underlying these effects remain poorly defined.

This gap was addressed by examining how the color-coded frame (green, yellow, or red), influences three key components of dietary decision-making (which will be referred to as decision domains from here on) at the neural level: WTP, healthiness perception, and tastiness perception. Previous research suggests that changes in healthiness perception are associated with emotional processing as reflected by amygdala activation [26], with similar results observed when comparing health labels to taste labels [27]. However, when participants were asked to rate healthiness perception on GDA labels (information based labels without color-codes) versus traffic light labels, which combine informational and color-coded elements and thus include an affective element, the findings indicated the involvement of analytical processing, reflected in activation of prefrontal and parietal cortices, and no involvement of the amygdala [28]. For tastiness perception the neural basis of FoPL-induced changes remains largely unexplored. Nonetheless, regions such as the middle orbitofrontal cortex and primary taste areas (anterior-dorsal insula and frontal operculum) are known to be involved in the integration of sensory and reward signals [29–33]. Based on these findings, we formulated the following neural hypotheses to test how color-coded frames modulate valuation and attribute-specific neural responses by including the ventral reward pathway and the dorsal control pathway.

**Hypothesis H2 (Ventral Reward Pathway)**

We hypothesize increased activation of the ventral reward pathway (vmPFC) across all three decision domains (WTP, healthiness, and tastiness). Specifically, green frames should be associated with stronger vmPFC activation compared to control trials. For tastiness perception, we additionally expect activation in sensory processing regions, including the insula [29–33].

**Hypothesis H3 (Cognitive Control Pathway)**

We hypothesize the engagement of the vmPFC and cognitive control pathway (dlPFC) for trials with a yellow and red frames compared to control trials. This pattern is predicted across all three decision domains, WTP, healthiness, and

tastiness, and reflects increased self-regulation and inhibitory control demands, in line with the neural FoPL framework [16,28,29]. Similar to H2, we hypothesize additional activation in the insula for tastiness perception.

## Methods

### Participants and data collection

A total of 51 participants were recruited for the study. Eligible participants had a Body Mass Index (BMI) between 18 and 25 [34], were 18–35 years old, fluent in German, raised in Germany to ensure familiarity with the food stimuli, and were right-handed. Right-handedness was confirmed using the Edinburgh Handedness Inventory [35]. Exclusion criteria included neurological or psychiatric disorders, standard MRI incompatibility, and adherence to a vegan diet as some depicted products contained non-vegan ingredients such as milk or eggs.

Following recruitment, further participants had to be excluded due to recalculated BMI values (n = 2), incidental MRI findings (n = 1), and technical issues (n = 8), resulting in 40 participants included in the final analysis. The final sample (28 females, 12 males) had a mean age of 23.8 years (SD = 3.1) and a mean BMI of 21.7 (SD = 1.7). All participants had normal or corrected-to-normal vision and fasted for four hours before the experiment, with hunger levels confirmed via self-report [36].

The inclusion of 40 participants was preregistered and informed by studies on methodological research on effect sizes in fMRI studies [15]. Given the event-related design of the current study, previous research has recommended a minimum of 20 participants to ensure reliable estimates [37]. In light of recent developments in the field [38,39], and the need to enhance both validity and generalizability, a sample size of 40 participants was deemed appropriate. This decision is further supported by recommendations from Geuter and colleagues (2018) [40], who suggest that a sample size of approximately 40 is sufficient to detect large effect sizes with high probability. In addition, the present study included a relatively high number of trials, 387 in total, which aligns with recent recommendations to consider trial count as a factor contributing to statistical efficiency in fMRI designs [41].

Participants were recruited primarily through the University of Bremen's mailing lists and advertisements. They received a base compensation of €7.50, with an additional €2.50 allocated for purchasing a food product in a Becker-DeGroot-Marshak (BDM) auction [42]. In the BDM auction, participants are incentivized to respond truthfully, as only one randomly selected trial is used for compensation. If a participant's bid "b" matched or exceeded a randomly selected price "n", they purchased the product at price "n", receiving €2.50 minus "n" plus the product. If "b"<"n", participants retained their €2.50. Thus, total compensation ranged from €7.50 to €10, plus one or two food products, depending on auction outcomes and the binary choice task (data not presented here). Before the experiment, participants received information on the Nutri-Score labels and provided written informed consent in accordance with the Declaration of Helsinki. The study was approved by the University of Bremen's ethics board. Data was collected between May and November 2023. The study was preregistered during data collection and thus before data analysis (for further details see S1 File).

### Experimental design

The study comprised two tasks within a within-subject repeated-measures design inside the scanner: a food product rating task (see Fig 1) and a binary choice task. The food product rating task was conducted during functional data acquisition, whereas the binary choice task took place during the anatomical scan; thus, only behavioral data were collected for the latter. Participants received instructions before the start of the experiment and again once they were lying in the fMRI scanner. Before each task, they were given the opportunity to complete practice trials while in the scanner. After the scanning session, participants completed a post-hoc questionnaire, re-rating all products on tastiness, healthiness, likeability, wanting, familiarity, WTP (with and without Nutri-Score), and were asked to give more information about their perceived

**A) Rating Task**

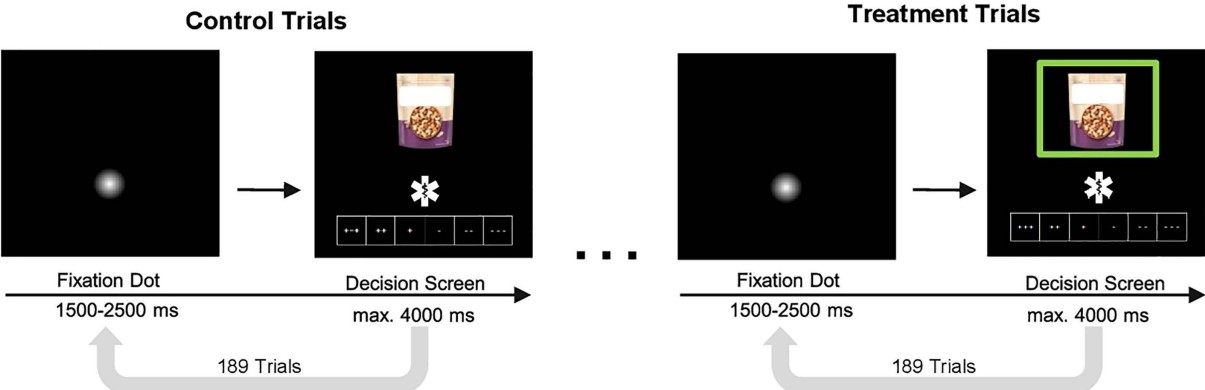

**B) Decision Screens in Rating Task**

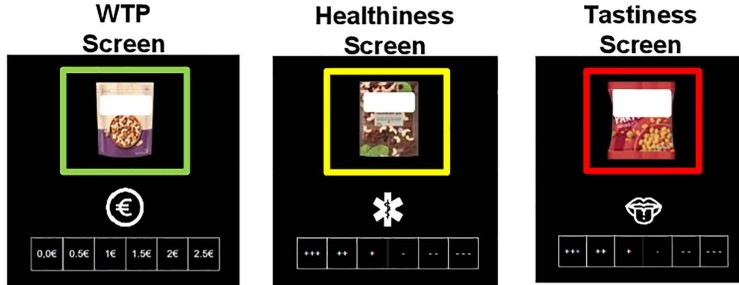

**Fig 1. Experimental design. A)** The figure shows one control condition trial and one treatment condition trial. All control condition trials were presented first, followed by the treatment condition trials in the same order. Each trial started with a fixation dot that lasts between 1500 and 2500 milliseconds (ms), with the dot positioned at eye-level, as is the decision-domain-icon. Thereafter, a decision screen was displayed for up to 4000 ms. Participants were instructed to make a decision according to the respective criteria by choosing one of six options via key press within the time frame. Once a decision was made, the chosen option was highlighted with a thicker frame for 300 ms. If participants exceeded the four-second time range, a screen displaying the text "You were too slow." appeared for 1000 ms before the next trial began. The rating task involved assessing 63 different products in a pseudo-randomized order. Each product was rated once in the no-frame condition and once in the frame condition, resulting in 189 trials for each condition. **B)** Each decision screen consisted of a product, a decision-domain-icon representing the ratings (WTP, healthiness, and tastiness) and the possible responses. Products were surrounded by a colored frame in green, yellow, or red, depending on their actual Nutri-Score. Due to copyright reasons products are edited in this figure and the original items are not shown.

stress. While completing the post-hoc questionnaire participants were allowed to eat the product they bought through the WTP-task. The data from the binary choice task are behavioral only and are not the focus of the current analysis; hence, they are not described further. Task presentation and behavioral data recording were conducted using Psychtoolbox extension Version 3.0.18 [43–45] on Octave Version 6.4 [46].

## Food product rating task

In the rating task (Fig 1A), participants completed 189 trials without the color-coded frame (control condition), followed by 189 trials with color-coded frames (treatment condition). In both conditions, they rated 63 different products on WTP, healthiness, or tastiness. To minimize learning effects, all participants started with the control condition. Each trial began with a fixation dot of variable length (1500–2500 ms), followed by a decision screen (≤ 4000 ms), where participants were asked to rate the products.

The 63 food products were categorized into seven groups: cereals, salty snacks, sweet snacks, yogurt, nuts, canned foods, and dips. Each category contained three products per Nutri-Score color level (green: A/B, yellow: C, red: D/E). The products were chosen based on a pretest to ensure familiarity (for further information see S2 File). Participants rated all 63 products in a pseudo-randomized order. The sequence was randomized once by the experimenter before the experiment and kept constant across all participants. Icons (Fig 1B) indicated the decision domain, asking participants to assess healthiness ('How healthy do you perceive this product to be?'), tastiness ('How tasty do you perceive this product to be?'), or WTP ('How much are you willing to pay for this product?'), which was based on the BDM-auction method [42]. Responses were made on a 6-point Likert scale, ranging from '+++' to '---' for healthiness and tastiness, and from €0 to €2.50 (in €0.50 increments) for WTP. The scale direction was counterbalanced across participants, such that the response options increased either from left to right or from right to left. Within each participant, however, the scale direction remained consistent throughout the experiment. Participants responded using two three-button response devices (one per hand), allowing for six possible inputs via the index, middle, and ring fingers of each hand. The leftmost button corresponded to the leftmost option on the screen, and the rightmost button to the rightmost option, ensuring spatial alignment between button positions and on-screen scale.

## Behavioral data analysis

Three repeated measures two-way ANOVAs were conducted to examine the effects of two within-subject factors, condition (control vs. treatment) and frame color (green, yellow, red),on the dependent variable, defined as the average choice per participant across all trials within the relevant category, for each decision domain (WTP, healthiness, and tastiness). Post-hoc t-tests with Bonferroni correction were used to explore significant differences. Missed trials were excluded from the analysis. The analyses were performed using R (version 4.1.2; [47] under RStudio (version 2023.6.2.561; [48]). Analysis of the response times can be found in the S3 File. An additional analysis comparing WTP responses from the rating task and the post-hoc questionnaire is provided in S4 File.

## fMRI data acquisition

Functional data were collected using a 3 Tesla Siemens MAGNETOM Vida Fit™ scanner equipped with a 64-channel head coil. A multi-band imaging sequence with a multi-band factor of 3 was used to optimize acquisition efficiency. The imaging parameters were as follows: repetition time (TR) = 1800 ms, echo time (TE) = 30 ms, flip angle (FA) = 77°, and field of view (FOV) = 192 × 192 × 138 mm³, with a voxel size of 2 × 2 × 2 mm, a slice thickness of 2 mm, and a total slice number of 69. For anatomical localization, T1-weighted MP-RAGE structural images were acquired with the following parameters: TR = 2100 ms, TE = 2.43 ms, FA = 8°, FOV = 256 x 256 x 208 mm³, with a voxel size resolution of 1 mm³, and an inversion time of 1100 ms.

## fMRI analysis

**fMRI preprocessing.** The analysis was conducted using SPM12 (version 7771) [49] on MATLAB Version 9.12 [50]. Slice-timing correction was applied with the first slice serving as the reference. Motion correction was performed, and the resulting six motion parameters were incorporated into the general linear model (GLM) as regressors of no interest. Additionally, motion estimates were examined to assess excessive head motion for potential participant exclusion (for further details, see S5 File). The anatomical images were normalized to the standard Montreal Neurological Institute (MNI152_2009) template in SPM and subsequently co-registered with the corresponding functional scans. Finally, spatial smoothing was applied using a 6 mm full-width at half maximum Gaussian kernel. A high-pass filter with a cutoff of 2160 s was applied to remove low-frequency drifts. To account for the temporal autocorrelation inherent in fMRI data, an autoregressive model of order 1 was employed. To estimate brain activation, a GLM analysis was conducted.

**General linear model analysis.** The GLM included 18 regressors of interest in addition to six motion regressors. The 18 regressors of interest were non-parametric, and time-locked to trial onsets, with response-dependent duration. Each decision domain (WTP, healthiness, tastiness) comprised six regressors: three for the treatment condition, where products were presented with color-coded frames (green, yellow, red), and three for the control condition, in which identical products were shown without a frame. Statistical inference was conducted using a family-wise error (FWE) cluster correction at $p < .05$, with an uncorrected two-sided voxel-level threshold of $p < .001$. To assess overlap across the decision domains (treatment > control condition) for each color-coded frame condition, null conjunction analyses were performed using a $p < .001$ threshold (two-sided) with 5% FWE cluster correction. The conjunction was implemented by computing the voxel-wise intersection of binarized statistical maps, retaining only voxels that were significant in all conditions [51]. For WTP, healthiness, and tastiness ratings, a single contrast was calculated between the decision periods in the control and treatment conditions for each color-coded frame condition. First-level contrasts were estimated using t-statistics in SPM at the whole-brain level on all individual subjects. This was followed by second-level group analyses via one-sample t-tests.

## Results

### Behavioral data- effects of the color-coded frame on WTP, healthiness perception, and tastiness perception

To address hypothesis H1, which proposed that green color-coded frames increase WTP, perceived healthiness, and perceived tastiness compared to control trials, while yellow and red frames decrease these measures, the analysis examined whether WTP, healthiness perception, and tastiness perception differed between control and treatment conditions at the behavioral level. Furthermore, it was tested whether the color of the frame modulated these effects, assessing whether different nutritional cues differentially influenced participants' decisions. Fig 2 illustrates the effects of the color-coded frame on WTP, healthiness, and tastiness, comparing treatment and control conditions for each color-coded frame condition.

The ANOVA for the WTP decision domain revealed significant main effects of treatment ($F(1,39) = 21.41$, $p < .001$, $\eta^2_{partial} = .35$) and color of the color-coded frame ($F(2,78) = 33.85$, $p < .001$, $\eta^2_{partial} = .46$). A significant treatment × color interaction ($F(2,78) = 12.69$, $p < .001$, $\eta^2_{partial} = .25$) indicated that the color-coded frame's impact varied by color. Post-hoc tests showed no WTP difference for green-labeled products ($M_{treatment} = €1.03$, $SD_{treatment} = €0.35$, $M_{control} = €1.04$, $SD_{control} = €0.33$, $t(39) = 0.30$, $p = .768$, $d = -0.05$), but a significant decrease for yellow ($M_{treatment} = €0.94$, $SD_{treatment} = €0.34$, $M_{control} = €1.03$, $SD_{control} = €0.34$, $t(39) = 3.96$, $p < .001$, $d = -0.63$) and red-labeled products ($M_{treatment} = €0.74$, $SD_{treatment} = €0.33$, $M_{control} = €0.91$, $SD_{control} = €0.31$, $t(39) = 6.28$, $p < .001$, $d = -0.99$).

While for healthiness perception the ANOVA revealed no significant main effect of treatment ($F(1,39) = 0.98$, $p = .328$, $\eta^2_{partial} = .02$), a strong main effect of color emerged ($F(2,78) = 221.10$, $p < .001$, $\eta^2_{partial} = .85$), along with a significant treatment effect × color interaction ($F(2,78) = 60.90$, $p < .001$, $\eta^2_{partial} = .61$). Green-labeled products received significantly higher healthiness ratings in the treatment condition ($M_{treatment} = 3.81$, $SD_{treatment} = 0.59$, $M_{control} = 3.35$, $SD_{control} = 0.46$, $t(39) = -5.78$, $p < .001$, $d = -0.91$), red-labeled products received lower ratings ($M_{treatment} = 2.12$, $SD_{treatment} = 0.45$, $M_{control} = 2.70$, $SD_{control} = 0.44$, $t(39) = 8.71$, $p < .001$, $d = -1.38$), while yellow-labeled products showed no difference ($M_{treatment} = 3.01$, $SD_{treatment} = 0.42$, $M_{control} = 3.01$, $SD_{control} = 0.46$, $t(39) = 0.05$, $p = .957$, $d = -0.01$).

The ANOVA for tastiness perception revealed that treatment ($F(1,39) = 22.73$, $p < .001$, $\eta^2_{partial} = .37$) and color ($F(2,78) = 21.60$, $p < .001$, $\eta^2_{partial} = .36$) both had significant main effects, with a significant treatment × color interaction ($F(2,78) = 11.16$, $p < .001$, $\eta^2_{partial} = .22$). Post-hoc tests revealed significantly lower tastiness ratings for yellow ($M_{treatment} = 3.87$, $SD_{treatment} = 0.50$, $M_{control} = 4.04$, $SD_{control} = 0.50$, $t(39) = 4.61$, $p < .001$, $d = -0.73$) and red-labeled products ($M_{treatment} = 3.55$, $SD_{treatment} = 0.62$, $M_{control} = 3.80$, $SD_{control} = 0.54$, $t(39) = 5.12$, $p < .001$, $d = -0.81$), but no difference for green-labeled products ($M_{treatment} = 4.03$, $SD_{treatment} = 0.48$, $M_{control} = 4.06$, $SD_{control} = 0.44$, $t(39) = 0.10$, $p = .922$, $d = -0.02$).

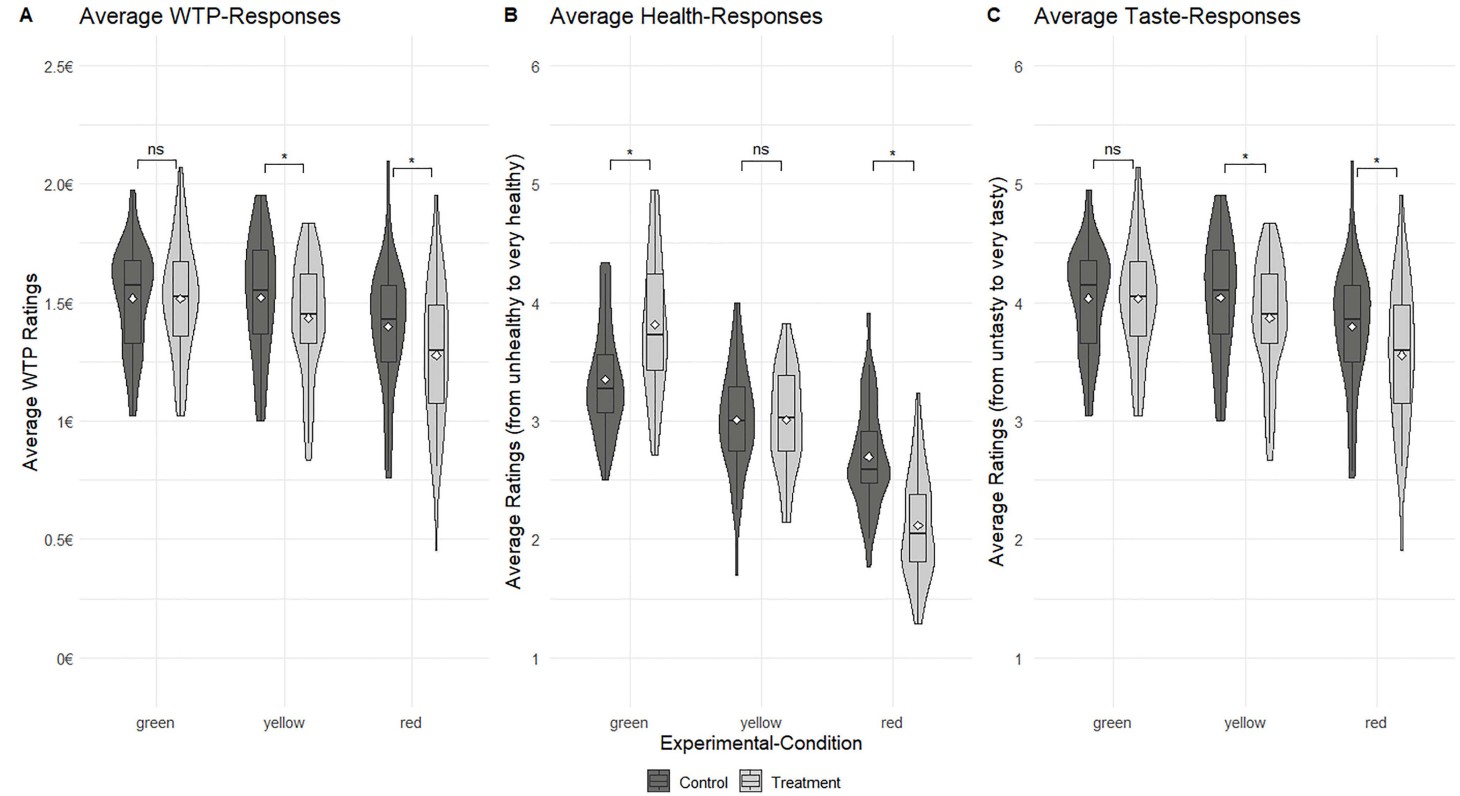

**Fig 2. Behavioral analysis of the rating task.** Violin plots presenting median and interquartile range for WTP (mean €), average healthiness ratings (-3(very unhealthy) to 3(very healthy)), and average tastiness ratings (-3(very untasty) to 3(very tasty)) in both the control condition and the treatment condition. Behavioral differences were analyzed using a paired t-test with Bonferroni correction. '***' indicates p < .001, and 'ns' denotes not significant. No p-values between .05 and .001 were observed.

### FMRI data

**General treatment effect (color-coded-frame).** First, an exploratory conjunction analysis was performed to assess the common effect of the color-coded frame across all decision domains (Fig 3; S1 Table).

Specifically, three contrasts (treatment > control for WTP, healthiness, and tastiness; for detailed results on these contrasts see S2–S4 Tables) were examined and overlapping regions of activation were identified. Significant clusters were observed in the dlPFC (bilateral; BA9), angular gyrus (BA39), superior temporal gyrus (STG, bilateral BA22), right anterior prefrontal cortex (aPFC; BA10), medial temporal gyrus (BA21), and left premotor cortex (BA6), among other regions (for abbreviations see figure legends).

Next, statistical differences in neural activation patterns between the treatment and the control conditions for each frame color and decision domain (WTP, healthiness, and tastiness) are reported.

**Green frame condition.** We hypothesized higher activation in the vmPFC across all decision domains when comparing green treatment condition with the respective control condition. Additionally, higher activation in the insula was hypothesized for tastiness perception. In the WTP condition, treatment trials elicited significantly higher activation in the aPFC (BA10), dlPFC (BA9), and STG (BA22), among other regions in comparison to control trials (see Fig 4 (orange overlay) and S5 Table). Thus, no activation was found in the hypothesized region (vmPFC) for WTP. For healthiness ratings, increased activation was observed in the vmPFC (BA11) as hypothesized (see Fig 4 (green

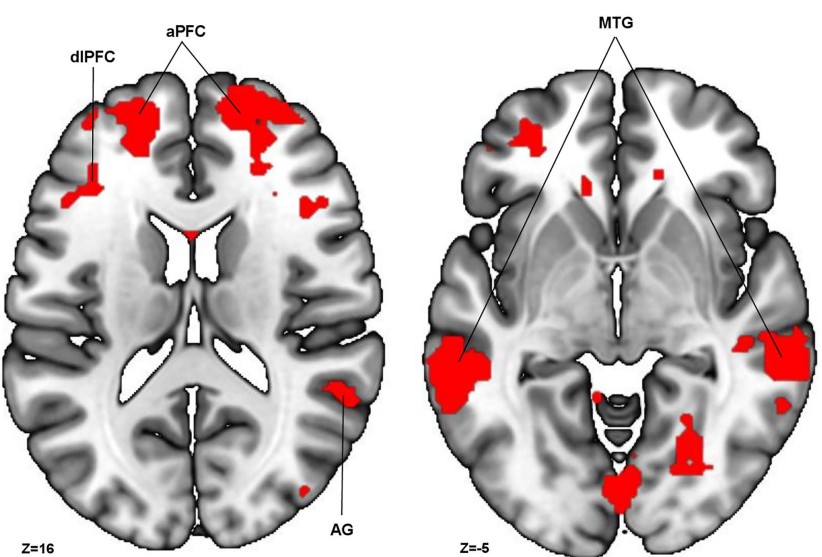

**Fig 3. Conjunction analysis of WTP, healthiness, and tastiness; treatment > control: T-map is projected onto an MNI_152_2009 template, p -threshold of <.001 and FWE cluster correction at p <.05 (df = [1, 39]).** Abbreviations in the figure: dlPFC = dorsolateral prefrontal cortex (BA46), aPFC = anterior prefrontal gyrus (BA10), AG = angular gyrus (BA39), MTG = medial temporal gyrus (BA21).

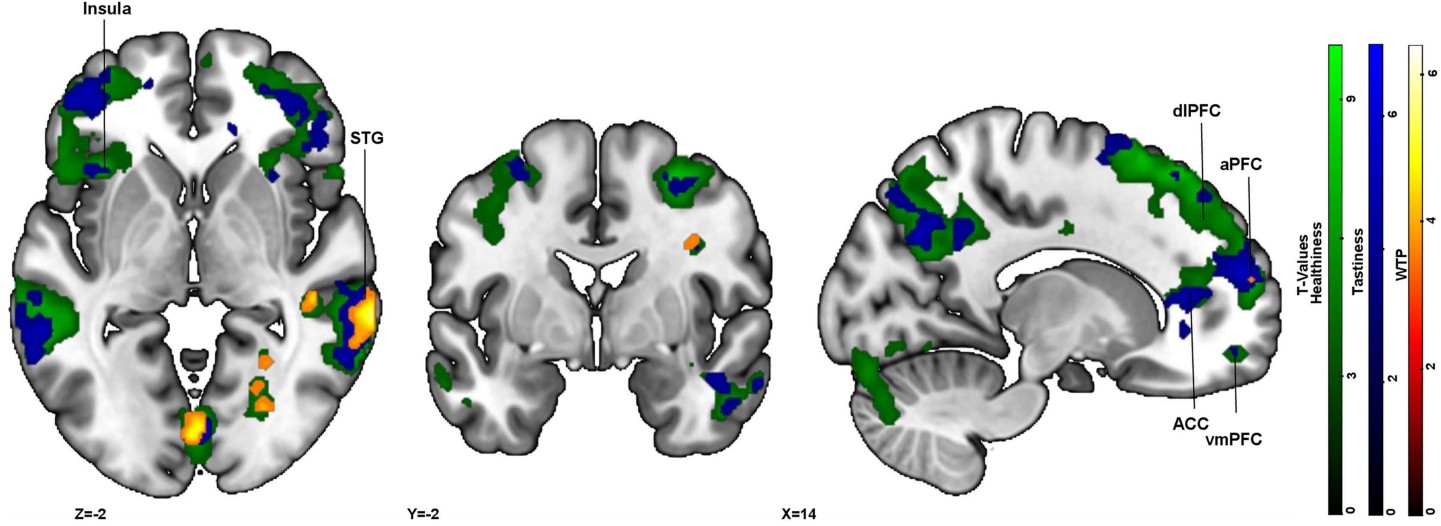

**Fig 4. Activation clusters derived from green treatment > green control condition for WTP (orange overlay), healthiness (green overlay), and tastiness (blue overlay): All t-maps are projected onto an MNI_152_2009 template, p <.001 and FWE cluster correction at p <.05 (df = [1, 39]) in the sequence first healthiness contrast, followed by tastiness contrast and then WTP contrast.** Abbreviations in the figure: ACC = anterior cingulate cortex (BA32), aPFC = anterior prefrontal cortex (BA10), dlPFC = dorsolateral prefrontal cortex (BA9), STG = superior temporal gyrus (BA22), vmPFC = ventromedial prefrontal cortex (BA11).

overlay) and S6 Table). Further higher activation were found in ACC (BA32), dlPFC (BA9) for healthiness perception. For tastiness ratings, higher activation in the hypothesized regions vmPFC and insula were found (see Fig 4 (blue overlay) and S7 Table). The activation pattern included further regions such as ACC, dlPFC, and STG in treatment

trials compared to control trials. All reported clusters were activated in only one direction, with no significant inverse effects detected.

**Yellow frame condition.** For the yellow frame condition, we hypothesized higher activation in vmPFC as well as dlPFC across all decision domains. For tastiness perception specifically, we further hypothesized additional activation in the insula activation compared to control condition. For WTP ratings, treatment trials elicited higher activation in the hypothesized (H3) dlPFC (see Fig 5 (orange overlay) and S8 Table) compared to control trials. The activation network further encompassed regions including aPFC (BA10), and STG (BA22). However, the hypothesized increase in vmPFC was not observed. For healthiness ratings, increased activation in the vmPFC (BA11) and dlPFC (BA46) was observed as hypothesized (H3), along with additional regions including the aPFC, ACC, thalamus, IFG, and STG (see Fig 5, green overlay, and S9 Table). Similarly, tastiness ratings in the treatment condition showed greater activation in the hypothesized regions vmPFC, dlPFC (BA46), and insula as well as the aPFC, IFG, and STG compared to control trials (see Fig 5, blue overlay, and S10 Table).

**Red frame condition.** The hypothesis for the red frame condition mirrored that of the yellow frame condition (H3). We expected increased activation in the vmPFC and dlPFC across all decision domains, with additional activation in the insula for tastiness perception. For WTP ratings, treatment trials showed significantly higher activation in the vmPFC (BA11), dlPFC (BA9), as hypothesized, as well as aPFC (BA10), insula, and thalamus (see Fig 6 (orange overlay) and S11 Table). For healthiness ratings, increased activation was observed in the vmPFC (BA11) and dlPFC (BA9), confirming our hypothesis (see Fig 6 (green overlay) and S12 Table). Additional activation was found in IFG (BA47), aPFC (BA10), and insula (BA13). Lastly, for tastiness ratings, the results also supported our hypothesis, showing an activation pattern involving the vmPFC (BA11), insula, and dlPFC (BA 9) (see Fig 6, blue overlay, and S13 Table). The activation network further included other regions such as the, aPFC (BA10), IFG, STG, and thalamus compared to control trials. Similar to

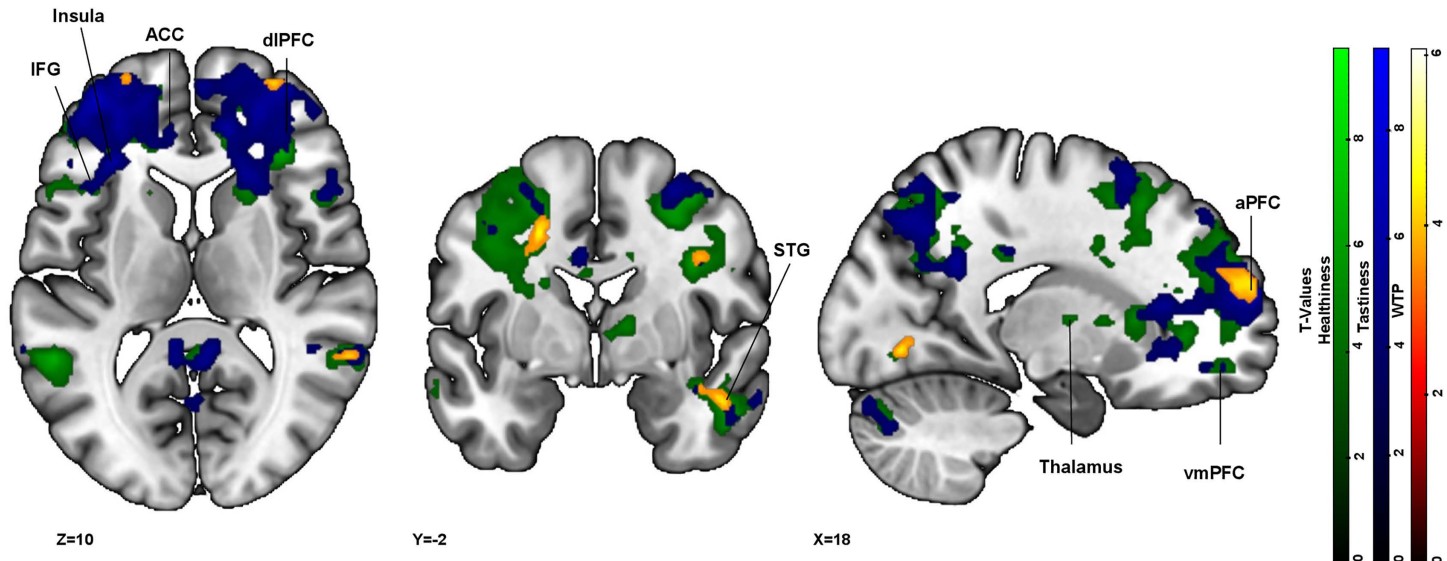

**Fig 5. Activation clusters derived from yellow treatment > yellow control condition for WTP (orange overlay), healthiness (green overlay), and tastiness (blue overlay): All t-maps are projected onto an MNI_152_2009 template, p < .001 and FWE cluster correction at p < .05 (df = [1, 39]) in the sequence first healthiness contrast, followed by tastiness contrast and then WTP contrast.** Abbreviations in the figure: ACC = anterior cingulate cortex (BA32), aPFC = anterior prefrontal cortex (BA10), dlPFC = dorsolateral prefrontal cortex (BA46), IFG = inferior frontal gyrus (BA47), STG = superior temporal gyrus (BA22), vmPFC = ventromedial prefrontal cortex (BA11).

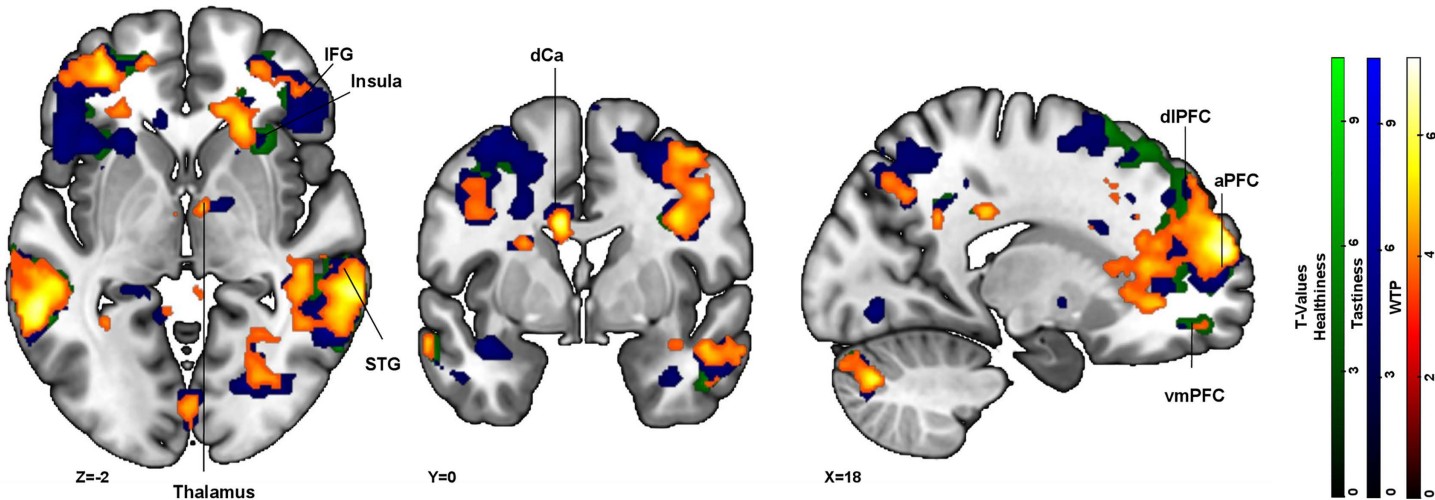

**Fig 6. Activation clusters derived from red treatment > red control condition for WTP (orange overlay), healthiness (green overlay), and tastiness (blue overlay): All t-maps are projected onto an MNI_152_2009 template in the sequence first healthiness contrast, followed by tastiness contrast and then WTP contrast.** A p -threshold of <.001 and FWE cluster correction at p < .05 (df = [1, 39]) was used. Abbreviations in the figure: aPFC = anterior prefrontal cortex (BA10), dCa = dorsal caudate, dlPFC = dorsolateral prefrontal cortex (BA9), IFG = inferior frontal gyrus (BA47), STG = superiortemporal gyrus (BA22), vmPFC = ventromedial prefrontal cortex (BA11).

the above-mentioned green frame condition and yellow frame condition analysis; no activation patterns in the opposite direction were detected.

## Discussion

### Behavioral data

This study hypothesized (H1) that a color-coded frame, representing a FoPL, influences decision values (WTP) and their corresponding attributes (healthiness and tastiness). The findings of the present study indicate that the color-coded frame predominantly functions to discourage unhealthy choices rather than to actively promote healthier alternatives. This pattern was observed both in the preregistered ANOVA and in the complementary multilevel analysis (sees S6 File). These results add to the mixed body of evidence on the impact of FoPLs on food choice behavior [14,19,52]. For example, one study reports that the Nutri-Score mainly encourages the selection of healthier products in binary choices [52], and another one finds an increase in purchase intention when a Nutri-Score "A" or Nutri-Score "B" is presented to participants but no significant difference when the Nutri-Score "C", "D", or "E" are shown to participants [14]. While other studies show a significant impact on the discouragement of unhealthy choices [2,20]. Further research is needed to clarify the sources of these heterogeneous effects, for example by examining the role of hunger state, participant's BMI, or differences in experimental setting. Additionally, the effects differ by frame color. It was hypothesized (H1) that color-coded frames would influence both valuation and attribute perception, with the green frame increasing perceived WTP, perceived healthiness, and perceived tastiness, and the yellow and red frames decreasing these decision domains. The results partially support this hypothesis. On a behavioral level, the green frame significantly increased healthiness perception, but contrary to expectations, it did not significantly enhance WTP or tastiness perception. This indicates that recognizing a product as healthier than initially thought, does not necessarily translate into higher WTP or an increase in tastiness perception. By contrast, the yellow frame significantly reduced both WTP and tastiness ratings, while leaving healthiness perception unaffected. This pattern suggests that an ambiguous or intermediate nutritional label may implicitly signal lower desirability,

potentially because consumers default to a binary classification system in which foods that are not explicitly labeled as "healthy", are considered as less subjectively valuable and less tasty. Finally, the red frame significantly reduced WTP, healthiness, and tastiness ratings, aligning with previous research [2,11,20]. This suggests that an external cue highlighting a product's unhealthiness, relative to an unlabeled version, reduces WTP by simultaneously diminishing tastiness perception. Importantly, the similarity of effects observed for WTP and tastiness ratings as opposed to healthiness perception further underscore the dominant role of taste perception in guiding food choices [32]. Overall, these findings suggest that the color-coded frame, which represents FoPLs, is more effective at discouraging unhealthy choices, by reducing perceived tastiness and subjective value, than at actively promoting healthier options.

### Implications of different activation patterns when contrasting treatment and control conditions

**Common activation patterns.** To examine how the color-coded frame alter the three decision domains at the neural level, common activation patterns, independent of frame color were analyzed. This exploratory analysis focused on the change in neural activation altered by the external cue in the treatment condition shared across all decision domains (WTP, healthiness perception, and tastiness perception). Compared to the control condition, the treatment condition elicited greater activation in the dlPFC (bilaterally; BA9) and aPFC (BA10), consistent with the neural FoPL framework [16]. The aPFC has been associated with goal-directed decision-making in humans [53], while the dlPFC plays a key role in self-regulation and cognitive control [54], suggesting that the color-coded frame engages neural mechanisms related to both controlled valuation and regulatory processes. Notably, these activation patterns are not limited to overall valuation (WTP), which has predominantly been the focus of prior research [16], but also influence the processing of specific decision attributes, including healthiness and tastiness perception.

In the following sections, the influence of the color-coded frame on the ventral reward pathway during valuation and on the dorsal control pathway during dietary decision-making is examined, with particular attention given to the differential color effects observed in the treatment conditions.

**The color-coded frame affects dietary decision-making through the valuation pathway.** The present paper investigated how the color-coded frame altered decisions in all three decision domains across the three frame conditions. Hypothesis H2 stated that the neural valuation pathway, specifically increased activation in the vmPFC, would emerge across all conditions and decision domains when compared to their respective control trials. The findings of this current study could only partially support our hypotheses. They suggest that the color-coded frames modulate valuation-related neural activity in all decision domains, however not across all color-coded frame conditions. Higher vmPFC activation was observed in the red WTP treatment condition compared to its control condition. This finding aligns with prior research indicating that external cues influence subjective valuation via the vmPFC [55,56]. However, our results did not show higher activation in the green and yellow color-coded condition for the WTP decision domain. Instead an activation pattern including aPFC (BA10), and ventral caudate were found, aligning with a recent meta-analysis on neural correlates of WTP [18]. Furthermore, we observed a correlation between WTP ratings and vmPFC activity (see S6 File). This finding indicates that, in addition to the average activation differences revealed by the color-code contrast analysis, interindividual variability in vmPFC responses is related to behavioral effect sizes. In both the healthiness and tastiness perception decision domains, higher vmPFC activation was observed across all color-coded conditions when compared to control trials, supporting Hypothesis H2 regarding the vmPFC's involvement in value computation. These findings support the hypothesis that the color-coded frame influences decision attributes by engaging valuation processes, as also shown in prior studies [27,33,56]. However, the results further suggest that the color-coded frames, representing an FoPL, act as either health-promoting or warning cues rather than uniform modifiers of subjective value, with behavioral effect size playing a key role in vmPFC engagement.

For tastiness perception, the red frame led to significantly greater activation in the right and left insula and vmPFC (treatment > control). Given the insula's established role in gustatory and interoceptive processing [32,57], and its

association with tastiness perception and expectations [58,59], these results suggest that health cues (the color-coded frame) might alter tastiness perception through vmPFC-insula interactions, modulating food valuation. Contrary to our hypothesis, vmPFC activation was not consistently observed across all conditions. Instead, for WTP in the treatment green and yellow conditions, greater activation in the ventral caudate and aPFC (BA10), regions involved in reward integration and goal-directed valuation [18,19], were observed. The ventral caudate is a well-established region in WTP decisions as shown in a recent meta-analysis by Newton-Fenner and colleagues (2023) [18], while the aPFC has been linked to goal-directed valuation and WTP representation [53,60,61]. The absence of consistent vmPFC activation, and the consistent dlPFC activation across all WTP conditions suggest that self-control mechanisms play a greater role in regulated dietary choices when an FoPL is shown. This is in line with prior research which has shown that regulated dietary decisions engage cognitive control regions (dlPFC) rather than hedonic valuation networks [53].

The present data suggest that vmPFC activation is not consistent within the different frame colors and the different decision domains. VmPFC activation was observed not only in WTP trials but also in healthiness and tastiness perception as hypothesized. Thus, a change in a decision attribute is also computed within the vmPFC. In traditional hierarchical models, it would be assumed that healthiness and tastiness perception, which are decision attributes, are processed separately before integration in the vmPFC [62,63]. A recent study challenges this by showing that decision attributes might be represented within the vmPFC, aligning with the findings of this study [63]. As this experiment is also based on the value-based decision making model [7], the present data complements these results by showing that a change in decision attributes is represented within higher activation in vmPFC.

**The color-coded frameengages cognitive control to modulate food valuation and choice.** Our third hypothesis (H3) stated that there seems to be an involvement of the dorsal control pathway, especially higher activation within the dlPFC, when the yellow and red color-coded frame were shown across all three decision domains. To test this hypothesis, the present study examined neural differences between treatment and control conditions across WTP, healthiness perception, and tastiness perception. Each frame color was analyzed separately, and a conjunction analysis was conducted to identify shared neural activation across all decision domains. Across all contrasts, greater activation in brain areas involved in self-regulation, particularly in the dlPFC were observed. Thus, our Hypothesis (H3) regarding the involvement of the dlPFC in all decision domains was supported for the yellow and red color-coded frame. Moreover, the data shows that the green color-coded frame integrates activation of brain regions linked to the cognitive control pathway as well. This activation pattern within the dlPFC plays a critical role in dietary self-control [54]. Increased dlPFC activation has been associated with healthier food choices [53] and predicts weight loss success in dieters [64,65]. Additionally, dlPFC grey matter volume predicts successful dietary regulation [66]. A meta-analysis further suggests that reduced dlPFC activation during self-control tasks is linked to higher BMI, underscoring its role in dietary regulation [67]. This is consistent with previous studies showing that dlPFC activation is associated with inhibitory control and goal-directed behavior [18,68]. Additionally, consistent greater inferior frontal gyrus (IFG; BA44) activation in treatment > control trials were observed across all conditions, supporting IFG's role in inhibitory control and resistance to food temptations when a FoPL is given [69].The IFG and dlPFC are key components of the dorsal control pathway, aligning with theoretical models of FoPLs as a cognitive intervention [16]. Moreover, greater activation in the ACC (treatment > control) was also observed in WTP-related contrasts, reinforcing its role in self-regulatory mechanisms and goal-directed behavior [18,68].

Importantly, activation in brain regions associated with self-control processes were not only related to WTP but also healthiness and tastiness perceptions. This suggests that the color-coded frame, which represents an FoPL, shape food perceptions by shifting cognitive processing toward self-regulation, which is also reflected by attribute-based evaluations. These findings support the notion that FoPLs function as regulatory nudges by engaging self-control mechanisms and encouraging a deliberate evaluation of food, rather than merely altering hedonic valuation or subjective value. The stronger recruitment of dlPFC, IFG, and ACC in the presence of the color-coded frames indicates that the color-coded frames impose a cognitive effort on taste-driven decisions by providing additional information, thereby suppressing hedonic

responses rather than automatically increasing health-driven valuations. These findings have real-world implications: if FoPLs primarily engage cognitive control rather than eliciting automatic valuation shifts, their effectiveness may depend on individual differences in self-regulatory capacity and impulsivity.

**The color-coded frame engages analytical processing in healthiness perception.** As part of Hypothesis 3 (H3), we expected stronger activation within the dorsal control pathway, particularly in the dlPFC, during healthiness perception in the yellow- and red-frame conditions. When comparing treatment and control conditions for healthiness perception across all frame colors (green, yellow, and red), we observed greater activation in the right and left dlPFC (BA46, BA9), the premotor cortex (BA6), and other regions. These findings align with previous evidence suggesting that FoPLs predominantly engage analytical processing when participants rate the healthiness of food products, as reflected by activation in prefrontal and parietal regions [28]. Although some studies have reported changes in amygdala activation [26,27], we did not observe significant differences in amygdala activation for healthiness perception. Taken together with the observed recruitment of cognitive-control regions, these results suggest that emotional engagement is not the primary mechanism through which color-coded frames affect healthiness perception. Nevertheless, this interpretation should be made with caution, as the amygdala is a small structure that is difficult to detect and localize in fMRI data. Additionally, interindividual variability in amygdala anatomy and potential inaccuracies introduced during spatial normalization may have reduced sensitivity to detect condition-specific effects.

**Beyond the hypothesized pathways.** Besides the hypothesized pathways mentioned above, we additionally observed activation in the thalamus when comparing the yellow treatment condition with its respective control condition for healthiness perception, and when comparing the red treatment condition with its respective control condition for tastiness perception and WTP. Furthermore, consistent activation in the superior temporal gyrus (STG) was found, as reflected both in the conjunction analysis and across all contrast analyses.

The thalamus plays a crucial role in integrating energy-state signals to guide dietary choices [17]. As a central hub within the homeostatic system, the thalamus modulates foraging and consumption behaviors based on internal physiological states [69,70]. In this study, increased activation in the thalamus for yellow-frame healthiness evaluations, as well as in red-frame trials for WTP and tastiness perception were observed. This pattern suggests that the color-coded frame may engage homeostatic mechanisms that regulate food valuation. Prior research indicates that homeostatic signals can modulate responses in primary taste regions to reduce perceived pleasantness and promote meal termination [32,71]. Similarly, the increased thalamic activation found in the present study when comparing treatment conditions with control conditions in the above-mentioned contrasts, may reflect a regulatory response. The color-coded frames prompt a shift toward reduced food reward, potentially aiding in limiting food consumption.

Increased activation in the STG in the treatment condition compared to the control condition across all decision domains (WTP, healthiness, and tastiness) were observed. Activation in the STG was evident in the conjunction analysis as well as in the analyses within each frame color (green, yellow, and red). STG activation is typically associated with social cognition [72]. However, the region is also recognized for its role in dietary decision-making: it contributes to processing the biological relevance of food and integrating contextual cues that shape desirability [73,74]. For instance, Charbonnier et al. (2015) [73] reported stronger STG activation when participants selected higher-calorie foods over lower-calorie alternatives, suggesting its involvement in contextual value computation during food selection. Furthermore, Manippa et al. (2017) [74] demonstrated a causal role of right STG activity in dietary decisions using transcranial random noise stimulation, showing that STG modulation influences high- vs. low-calorie food choices. Nevertheless, the direct involvement of the STG in food choices affected by nutritional labels remains unclear.

## Limitations of the study

Due to experimental purposes and copyright reasons the original Nutri-Score was simplified by reducing the original five color scale to three categories [75,76]. Although this method allowed for greater experimental control, facilitated data

analysis and interpretation, it may not fully represent the label's real-world sophistication. In real-world settings, the Nutri-Score label is typically smaller and embedded directly in the packaging, making it less visually salient than in the present task. From a theoretical perspective, FoPLs are typically classified as cognitive nudges, as they provide information that requires deliberate processing [2,77]. However, the Nutri-Score also includes affective elements through its color-coded grading system, which triggers intuitive associations (green = "good", red = "bad") and can influence behavior via affect-driven mechanisms [2,78]. In the present study, the manipulation specifically emphasized this affective component, just as with the actual Nutri-Score, where the letter often does not need to be read once the color code is understood, participants could rely solely on the color-coded frame to infer a product's nutritional quality. Prior research shows that red color cues, both in background [79], and ambient settings [80], affect food decisions and promote avoidance of unhealthy choices, suggesting that the frame likely influenced decisions through an affective mechanism. To verify that the evaluative meaning of the frames was preserved, we compared WTP ratings from the fMRI task with ratings from a post-hoc questionnaire across treatment and color-code conditions (see S4 File). No significant differences were found, except for the green frame condition, likely because participants had broken their fast during the questionnaire. Comparable findings were reported by Enax and colleagues (2015) [19]. These results confirm that participants inferred the intended nutritional information even without the letter grade, indicating that our manipulation preserved the cognitive content of the Nutri-Score while making its affective, color-based cue more salient.

In the present study, all participants completed the control trials before the treatment trials, following a fixed trial sequence. This order was intentionally chosen to ensure that the control condition remains unaffected by prior exposure to the color-coded frame used in the treatment condition. While this decision was made to reduce the bias in interpretation of the control data, we acknowledge that it might introduced order effects such limiting the degree of randomization. Specifically, prior exposure to the same products and decision-domain icons during the control phase may have influenced participants' responses in the treatment phase. The use of a constant trial sequence across participants was chosen to meet constraints inherent to the event-related fMRI design, such as ensuring sufficient temporal separation between conditions for accurate modeling of condition-specific neural activity. Although we applied a pseudorandomization procedure to avoid presenting trials of the same condition consecutively, full randomization would still have been possible within the same constraints, and our approach may therefore have introduced order-related confounds. Future studies might address these limitations by employing a between-subjects design, counterbalancing the order of conditions, or using a within-subject design with counterbalanced condition orders across participants. Additionally, varying the trial sequence between participants could help mitigate potential confounds and enhance the robustness of the findings. However, these between subject approaches comes with trade-offs, such as the inability to estimate within-subject treatment effects, the requirement for larger sample sizes, or the difficulty of establishing a clean control condition.

A further important limitation of this study is that it focuses solely on single-shot decisions, without examining the long-term effects of external cues on dietary choices. It remains unclear whether the reliance on self-control in decision-making is stable over time, whether it increases or decreases with repeated exposure to such cues, or how social factors influence food choices affected by such labels. Furthermore, reward learning plays a crucial role in shaping dietary behaviors, yet its influence on long-term food valuation is not addressed in the current framework. For instance, while a FoPL may initially promote the selection of a nutritionally optimal product, the consumer's willingness to repurchase the item may ultimately depend on consumption and post-consumption experiences. If a product is perceived as less palatable than expected, its reward value may decrease, leading to a lower likelihood of future purchases.

Additionally, participants fasted for four hours before the experiment, a factor that might impact both behavioral and neural outcomes. Since rating occurred in a fasted state, decisions may have been more impulsive, mirroring food purchasing under hunger conditions. The fasted state and the length of the study might have also affected the fatigue of participants during the experiment. Moreover, participants' motivation and health knowledge were not assessed, and the

sample was neither gender-balanced nor representative of a broader population, as it consisted solely of individuals aged 18–35 [81], which limits the generalizability of the findings.

## Outlook

FoPLs may have differential effects across age groups, including adolescents and older adults, as compared to young adults (18–35 years). Furthermore, factors like education and budget for food purchases can alter the effects of the FoPL [16]. Given that FoPLs are designed for broad consumer populations, future studies should examine age-related differences, education, and economic background in their influence on dietary decision-making. The findings highlight potential areas for future research on FoPLs and food decision-making. First, this study included only participants with a normal BMI, yet obesity and eating disorders are known to alter reward processing and decision-making [82]. For instance, individuals with obesity exhibit steeper delay discounting for food rewards, accompanied by altered activation in the dorsal striatum and anterior cingulate cortex as has been demonstrated in a recent study of this study's research group [83]. Future research should examine whether real FoPL effects differ across weight groups, particularly in populations with heightened reward sensitivity to food cues. Additionally, individual differences in motivation, health knowledge, and time pressure, among others, may shape the influence of FoPLs on decision-making [16]. Identifying how these contextual and cognitive factors interact with labeling interventions could provide insights into personalized strategies for improving dietary choices. By addressing these individual and contextual factors, future studies could refine the understanding of how FoPLs alter dietary decisions across diverse populations.

## Conclusion

The present study investigated the neural mechanisms through which color-coded frames, serving as a proxy for FoPL, shape dietary decisions by disentangling key decision-making components: WTP, perceived tastiness, and perceived healthiness. By integrating a value-based decision-making framework [7] with recent advances in FoPL research [16], the findings confirm that color-coded nutritional cues systematically influence consumer behavior. In line with previous research, a significant decrease in WTP for products with red and yellow frames was observed in the current study, reinforcing evidence that FoPLs modulate the subjective value of products. Furthermore, the results confirm that the frame manipulation which served as a proxy for a FoPL effectively manipulate healthiness perception, supporting their intended function as a nudge toward healthier choices. Additionally, the current study shows that tastiness perception is also influenced by the color-coded frame, revealing a mechanism through which health labels shape dietary decisions. Although food decisions are often automatic or habitually primed, the findings suggest that the color-coded frame, and thus FoPL promote more deliberate and goal-directed decisions by shifting food valuation processes. This is underlined by the findings that on the neural level, color-coded frame-induced changes in valuation were reflected in activation patterns across key decision-making-related brain regions. Increased engagement of the vmPFC, a core region in the reward pathway, the dlPFC, implicated in cognitive control, and the thalamus, a key structure in homeostatic regulation, was observed. Specifically, changes in WTP were associated with vmPFC activation, reinforcing its role in subjective value computation. Healthiness perception was linked to differential recruitment of the vmPFC and dlPFC, suggesting interplay between reward valuation and cognitive control when evaluating food healthiness. Notably, changes in tastiness perception were accompanied by activation in the insula, a key sensory processing region, alongside the vmPFC, indicating a direct neural link between taste-related valuation and sensory integration. Together, these findings suggest that FoPLs influence dietary decisions by recruiting both reward and control networks, integrating homeostatic processes to influence food valuation.

## Supporting information

**S1 File. Preregistration.**
(DOCX)

**S2 File. Stimuli.**
(DOCX)

**S3 File. Response time analysis.**
(DOCX)

**S4 File. Post-hoc questionnaire analysis.**
(DOCX)

**S5 File. Motion detection estimation.**
(DOCX)

**S6 File. Multilevel analysis of behavioral data.**
(DOCX)

**S7 File. Parametric modulation analysis with WTP.**
(DOCX)

**S1 Table. Conjunction of the contrasts according to a logical "and" calculation: WTP (treatment > control), healthiness (treatment > control), tastiness (treatment > control).**
(DOCX)

**S2 Table. Brain regions showing significant activation for the contrast treatment > control during WTP ratings.**
(DOCX)

**S3 Table. Brain regions showing significant activation for the contrast treatment > control during healthiness ratings.**
(DOCX)

**S4 Table. Brain regions showing significant activation for the contrast treatment > control during tastiness ratings.**
(DOCX)

**S5 Table. Brain regions showing significant activation in treatment > control (green frame condition) during WTP ratings.**
(DOCX)

**S6 Table. Brain regions showing significant activation in treatment > control (green frame condition) during healthiness ratings.**
(DOCX)

**S7 Table. Brain regions showing significant activation in treatment > control (green frame condition) during tastiness ratings.**
(DOCX)

**S8 Table. Brain regions showing significant activation in treatment > control (yellow frame condition) during WTP ratings.**
(DOCX)

**S9 Table. Brain regions showing significant activation in treatment > control (yellow frame condition) during healthiness ratings.**
(DOCX)

**S10 Table. Brain regions showing significant activation in treatment > control (yellow frame condition) during tastiness ratings.**
(DOCX)

**S11 Table. Brain regions showing significant activation in treatment > control (red frame condition) during WTP ratings.**
(DOCX)

**S12 Table. Brain regions showing significant activation in treatment > control (red frame condition) during healthiness ratings.**
(DOCX)

**S13 Table. Brain regions showing significant activation in treatment > control (red frame condition) during tastiness ratings.**
(DOCX)

## Acknowledgments

We thank Nourat Noemi Alazza for her support during the data collection and Katharina Tucholski for her assistance in proofreading the manuscript. We additionally would like to thank Prof. Dr. Nora Szech, who provided many valuable suggestions for the conception of the study, but due to her sudden death did not have the opportunity to get to know the results of her fruitful contributions

## Author contributions

**Conceptualization:** Negin Javaheri, Niels Doehring, Manfred Herrmann.

**Data curation:** Negin Javaheri, Peter Erhard.

**Formal analysis:** Negin Javaheri, Niels Doehring, Revati Mulay, Peter Erhard.

**Funding acquisition:** Manfred Herrmann.

**Investigation:** Negin Javaheri, Peter Erhard.

**Methodology:** Negin Javaheri, Niels Doehring, Peter Erhard.

**Project administration:** Negin Javaheri, Niels Doehring, Manfred Herrmann.

**Resources:** Manfred Herrmann.

**Software:** Negin Javaheri, Niels Doehring, Revati Mulay, Peter Erhard.

**Supervision:** Manfred Herrmann.

**Validation:** Negin Javaheri, Niels Doehring, Peter Erhard.

**Visualization:** Negin Javaheri.

**Writing – original draft:** Negin Javaheri.

**Writing – review & editing:** Negin Javaheri, Niels Doehring, Revati Mulay, Peter Erhard, Manfred Herrmann.

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
