## [Decision Letter · Decision Letter 0]

4 Jul 2025

Dear Dr. Javaheri,

Thank you for submitting your manuscript to PLOS ONE. After careful consideration, we feel that it has merit but does not fully meet PLOS ONE’s publication criteria as it currently stands. Therefore, we invite you to submit a revised version of the manuscript that addresses the points raised during the review process.

We look forward to receiving your revised manuscript.

Kind regards,

Akitoshi Ogawa, Ph.D.

Academic Editor

PLOS ONE

**Journal Requirements:**

1. When submitting your revision, we need you to address these additional requirements. Please ensure that your manuscript meets PLOS ONE's style requirements, including those for file naming. The PLOS ONE style templates can be found at https://journals.plos.org/plosone/s/file?id=wjVg/PLOSOne_formatting_sample_main_body.pdf and https://journals.plos.org/plosone/s/file?id=ba62/PLOSOne_formatting_sample_title_authors_affiliations.pdf 2. Thank you for stating the following financial disclosure: This work was funded by Deutsche Forschungsgemeinschaft (DFG, German Research Foundation)-GRK2739/1-Project Nr. 447089431-Research Training Group: KD2 School-Designing Adaptive Systems for Economic Decisions   Please state what role the funders took in the study.  If the funders had no role, please state: "The funders had no role in study design, data collection and analysis, decision to publish, or preparation of the manuscript." If this statement is not correct you must amend it as needed. Please include this amended Role of Funder statement in your cover letter; we will change the online submission form on your behalf. 3. Please note that your Data Availability Statement is currently missing the repository name and/or the DOI/accession number of each dataset and a direct link to access each database. If your manuscript is accepted for publication, you will be asked to provide these details on a very short timeline. We therefore suggest that you provide this information now, though we will not hold up the peer review process if you are unable. 4. When completing the data availability statement of the submission form, you indicated that you will make your data available on acceptance. We strongly recommend all authors decide on a data sharing plan before acceptance, as the process can be lengthy and hold up publication timelines. Please note that, though access restrictions are acceptable now, your entire data will need to be made freely accessible if your manuscript is accepted for publication. This policy applies to all data except where public deposition would breach compliance with the protocol approved by your research ethics board. If you are unable to adhere to our open data policy, please kindly revise your statement to explain your reasoning and we will seek the editor's input on an exemption. Please be assured that, once you have provided your new statement, the assessment of your exemption will not hold up the peer review process.

Reviewers' comments:

Reviewer's Responses to Questions

**Comments to the Author**

1. Is the manuscript technically sound, and do the data support the conclusions?

Reviewer #1: Yes

Reviewer #2: No

2. Has the statistical analysis been performed appropriately and rigorously?

Reviewer #1: Yes

Reviewer #2: No

3. Have the authors made all data underlying the findings in their manuscript fully available?

Reviewer #1: No

Reviewer #2: Yes

4. Is the manuscript presented in an intelligible fashion and written in standard English?

Reviewer #1: Yes

Reviewer #2: Yes

**Reviewer #1:**  The present study examined how package labels influence food valuation—specifically perceived healthiness, perceived tastiness, and willingness to pay (WTP). The authors demonstrate that color‐coded frames indicating Nutri‐Score categories modulate these ratings as well as WTP. These behavioral changes are accompanied by increased fMRI BOLD responses in reward‐related, cognitive-control, and homeostatic regions. Overall, the manuscript clearly states its aims, results, and conclusions. My comments are as follows.

Fixed order of conditions:

The control trials always preceded the treatment trials. This fixed sequence could be a critical confound: differences in behavior and neural activity may reflect order effects rather than the manipulation itself. Please discuss this issue.

Visual disparity between conditions:

The treatment condition included a colored frame around each food image, whereas the control condition did not. Consequently, any behavioral or neural differences might stem from the mere presence of salient color cues rather than their Nutri-Score meaning. Please consider addressing this alternative explanation.

vmPFC/striatum correlations with WTP:

BOLD activity in reward-processing areas such as the ventromedial prefrontal cortex and striatum is consistently correlated with WTP. Did your data replicate this established relationship? Reporting such a validation would strengthen confidence in the fMRI findings.

Sample-size justification:

The manuscript does not explain how the sample size was determined. Please provide a power analysis or other rationale (e.g., prior literature, resource constraints) to justify the chosen N.

**Reviewer #2:**  1. Page 6: I am a bit surprised that you have not formulated any hypotheses, especially for the food rating task (but also for the fMRI experiment). It would be interesting to be able to compare your findings to what you would expect based on the literature.

2. Page 8, lines 162-164: Why are the behavioral data not included in the paper? It might be interesting to see whether what happens on a neural level is in line with what participants indicate themselves.

3. Page 9, lines 179-181: Based on the title, I initially expected the study to focus specifically on the Nutri-Score front-of-pack (FOP) label. However, the experimental design uses only three colors derived from the Nutri-Score system, without incorporating its full structure—most notably, the accompanying letter grades and the complete five-color scale. These additional components are central to the Nutri-Score’s intended function, which is to avoid overly simplistic, dichotomous thinking and to provide a more nuanced, graded evaluation of product healthfulness. From a theoretical standpoint, and in line with Cadario and Chandon’s (2020) typology, the Nutri-Score is best classified as a cognitive nudge, as it provides consumers with explicit evaluative information. In contrast, the manipulation tested in your study—color-coded product frames—would more accurately be described as an affective nudge, since it relies on intuitive and emotional responses to color cues rather than direct informational content. While the neural findings you report are potentially valuable, I believe the current framing of the paper as an investigation of the neural correlates of a front-of-pack label may be misleading. A more precise positioning would clarify that the study focuses on an affective cue inspired by elements of FOP labeling, rather than a direct examination of a complete label system such as the Nutri-Score. In my opinion, the experiment does not answer the question raised in the introduction: “How do these labels alter food perception and decision-making?”.

4. Following up on the previous comment, the theoretical background should elaborate on how the chosen nudge (in your case the use of colored frames) has affected consumers’ perceptions. Currently, there is very limited information on how the Nutri-Score affects WTP, healthiness perceptions and tastiness perceptions and the focus is immediately on neural activation. As I would no longer focus on the Nutri-Score in your research, I do not suggest to include these findings in the theoretical background of the paper. However, I would recommend to include some overview of how affective nudges have impacted WTP, healthiness perceptions and tastiness perceptions in previous research.

5. I think it is important to include the fact that you ran a pretest in the paper to select the different products (which is now explained in the S2 file). Based on this pretest, I am bit worried about the 6 options you used to measure willingness to pay, given that some products were priced higher than the maximum (i.e., €3.19). Maybe this is something to include in the limitations section of your study?

6. Although the fMRI sample size (n = 40) is solid, the behavioral sample may be underpowered. A formal power analysis would help justify the adequacy of this sample.

7. Page 9, lines 184-185: Although I understand why you first showed the control trials, I would have suggested to counterbalance across participants and present the control trials first to some participants and the experimental trials first to others. You now introduce order confounds. This should be included as a limitation of your study.

8. Page 10, lines 193-195: I do not understand why all participants rated the products in the same sequence. This design choice introduces potential order effects and undermines randomization.

9. Page 13-14: Your current analyses do not take into account the hierarchical data structure (i.e., you average the responses across the 21 green, yellow, and red-framed products) for each dependent variable. Hence, you ignore trial level variability as well as potential differences between stimuli. Therefore, I would recommend to run multilevel analyses. In addition, multilevel models can deal better with missing data (which you very likely have as respondents have to answer quickly).

10. Page 13-14: In line with comment number 5, I am worried that the fact that you found that WTP and tastiness dropped in the color conditions may be partly explained by participant fatigue.

11. Page 13-14: I would suggest reporting effect sizes (cohen’s d) for the post hoc tests.

12. It is important to rewrite the discussion. Now it is constantly mentioned that you studied whether labels impact decision values, this claim overstates the manipulation. A color-coded frame is not equivalent to a front-of-pack nutrition label.

13. Page 19, lines 383: You mention “it was hypothesized” though no hypothesis were included in the manuscript.

14. Page 19, lines 387-388: Why would you expect that the green frame would positively impact tastiness perceptions? Participants were informed upfront that the frames were health-related, so why would participants infer that a healthier product is also tastier? This is actually going against a large body of literature on the unhealthy = tasty intuition (Raghunathan, Naylor, & Hoyer, 2006). Reference: Raghunathan, R., Naylor, R. W., & Hoyer, W. D. (2006). The unhealthy= tasty intuition and its effects on taste inferences, enjoyment, and choice of food products. Journal of Marketing, 70(4), 170-184.

Some smaller comments:

1. Page 13, line 173: I would start the sentence with “An ANOVA revealed…”

2. Small typo on page 14, line 274: η²partial = 46 should be η²partial = .46

**Do you want your identity to be public for this peer review?** For information about this choice, including consent withdrawal, please see our Privacy Policy

Reviewer #1: No

Reviewer #2: No

---

## [Author Response · Author response to Decision Letter 1]

18 Aug 2025

From,

Negin Javaheri

Department of Neuropsychology and Behavioral Neurobiology,

University of Bremen

To,

Dr. Ogawa

Subject: Rebuttal Letter for Manuscript PONE-D-25-19520

From attributes to value: neural correlates of a front-of-package label on food decision-making – an fMRI study

PLOS ONE

Dear Dr. Ogawa,

We sincerely appreciate the invitation to revise and resubmit our Manuscript PONE-D-25-19520entitled " From attributes to value: neural correlates of a front-of-package label on food decision-making – an fMRI study" to PLoS One. We are grateful for the valuable comments provided by you, as well as both of the reviewers that helped us to clarify and improve the quality of our work.

Below, we will provide a detailed response to each of the comments. For the sake of clarity, the journal requirements and the reviewers’ comments are included in bold, followed by our responses in plain text. All changes made to the manuscript are attached as ‘Revised_Manuscript_PONE-D-25-19520_with_Track_Changes’, further we attach the manuscript without track mode as ‘Revised_Manuscript_ PONE-D-25-19520’. Additionally, we revised the Discussion section (see page 23, line 515 ff.). Please note that the page and line references are based on the revised manuscript with track changes.

Thank you for your time.

Best Regards,

Negin Javaheri

Journal Requirements:

1. When submitting your revision, we need you to address these additional requirements. Please ensure that your manuscript meets PLOS ONE's style requirements, including those for file naming.

We apologize for the inconsistencies in the submission. We have revised the manuscript according to the requirements.

2. Thank you for stating the following financial disclosure: This work was funded by Deutsche Forschungsgemeinschaft (DFG, German Research Foundation)-GRK2739/1-Project Nr. 447089431-Research Training Group: KD2 School-Designing Adaptive Systems for Economic Decisions. Please state what role the funders took in the study. If the funders had no role, please state: "The funders had no role in study design, data collection and analysis, decision to publish, or preparation of the manuscript." If this statement is not correct you must amend it as needed. Please include this amended Role of Funder statement in your cover letter; we will change the online submission form on your behalf.

We thank for the additional indication. We have extended the sentence and deleted in the revised manuscript (see page 34, line 849 ff.)

3. Please note that your Data Availability Statement is currently missing the repository name and/or the DOI/accession number of each dataset and a direct link to access each database. If your manuscript is accepted for publication, you will be asked to provide these details on a very short timeline. We therefore suggest that you provide this information now, though we will not hold up the peer review process if you are unable.

As we are commited to principles of open science, we will be making the data available under OSF. The DOI will be given when the manuscript will be accepted. Currently, we are preparing the data for publication, e.g. defacing fMRI data for anonymization.

We appreciate this suggestion and have made a data sharing plan. We will be uploading the MRI data in OSF, once we finished the defacing to ensure anonymity. Additionally, we will upload all analysis codes, and all behavioral data set both for studies referred to in the present manuscript as well as piloting and other data mentioned here.

Reviewer#1

1. Fixed order of conditions: The control trials always preceded the treatment trials. This fixed sequence could be a critical confound: differences in behavior and neural activity may reflect order effects rather than the manipulation itself. Please discuss this issue.

We thank the reviewer for highlighting this important concern, which is also shared by the reviewer 2 (point 7). We fully acknowledge that presenting the control trials before the treatment trials may have introduced order effects, potentially confounding behavioral and neural differences. We have now addressed this point explicitly in the Limitations section. Our rationale for using a fixed order was to avoid potential carryover effects from the treatment condition onto the control trials. Presenting the treatment condition first might have biased participants' baseline responses, undermining the interpretability of the control condition. Nevertheless, we agree that the fixed sequence limits the ability to dissociate treatment effects from possible order effects. We now discuss this possible confound in the Limitations section (see page 29, line 720 ff.).

2. Visual disparity between conditions: The treatment condition included a colored frame around each food image, whereas the control condition did not. Consequently, any behavioral or neural differences might stem from the mere presence of salient color cues rather than their Nutri-Score meaning. Please consider addressing this alternative explanation.

We acknowledge the reviewer’s concern that the color-coded frame used in the treatment condition may have served as a salient visual cue rather than being interpreted as meaningful information on healthiness of food components. This issue was also raised by Reviewer 2 (see point 3) has been discussed within our research team, and was subject on various analyses and pilot studies not mentioned in the present manuscript. We would like to stretch the various reasons behind using a color-coded frame and not just the Nutri-Score label including a colored scheme as well. From the neuroeconomics literature, we know that attention strongly shapes decision-making. Fehr & Rangel (2011), include attention as one of the main factors that control decisions. Further, from the work by Reeck and colleagues (2017), we know that manipulation of attention can alter overall choice. With respect to experimental reasons The here used color-coded frame allowed us to control for how much attention participants gave to the treatment, as on a visual level there was no escape from the frame. If a label was used, then the frame could have been ignored, which would also mean that our treatment might have not worked. In the following, we show in two different analyses that corroborates our view that the frame was seen as more than a color cue and participants perceived it as a frame containing information of the healthiness of the product’s nutritional components.

As a first measure to assess this issue whether the color-coded frame was interpreted as nutritional information rather than merely a salient visual cue, we analyzed participants' responses from a post-experiment questionnaire. All participants were asked to re-rate the products in a post-hoc questionnaire, answering the question how much they are willing to pay for the product(“ How much are you willing to pay for the product”), and how much they are willing to pay for the product given their real Nutri-Score (“ How much are you willing to pay for the product (Nutri-Score X)”) on a continuous scale. We than compared the average WTP of participants in the fMRI and in the post-hoc questionnaire to investigate whether average responses differed between the MRI setting and in the post-hoc questionnaire. A repeated-measure ANOVA was conducted using average responses of participants per setting, treatment condition, and color-coded frame. A Shapiro–Wilk test indicated that all variables were normally distributed (all p-values > 0.1), and no outliers were identified. The results of an ANOVA showed a significant main effect of setting (F (1, 39) = 8.84, p = .005, η² = .040), main effect of treatment (F (1, 39) = 28.57, p < .001, η² = .008), and main effect of the color-code (F (2, 78) = 56.84, p < .001, η² = .094). The finding was interpreted in a way that WTP differed between settings, between treatment, and color-codes, with a significant interaction effect between setting and treatment (F (1, 39) = 6.63, p = .014, η² = .002), as well as the interaction between treatment and color-coded frame (F (2, 78) = 52.13, p < .001, η² = .019). Further, the three-way interaction between setting, treatment condition, and color-coded frame became also significant (F (2, 78) = 7.75, p = .001, η² = .002), indicating that the combined influence of setting and treatment condition varied across color-coded frames. In addition, post-hoc paired t-tests were conducted between the settings within each color-coded frame and treatment condition with Bonferroni correction . For average WTP per setting, treatment conditions and color-coded frame conditions, see the following Figure 1.

Figure 1. Average WTP per Setting, Treatment and Color-Coded Frame Condition

In both the control condition as well as in the treatment condition, average WTP was significantly higher in the post-hoc-questionnaire setting (Mpost-hoc-control = 1.09, SDpost-hoc-control = 0.31; Mpost-hoc-treatment = 1.06, SDpost-hoc-treatment = 0.37 ) compared to the MRI setting (MMRI-control = 0.99, SDMRI-control = 0.32; pBonf < .05; MMRI-treatment = 0.90, SDMRI-treatment = 0.36; pBonf. < .001). For the green color-coded frame condition, a significant difference in WTP between settings emerged both in the control group (pBonf.- < .05) and in the treatment group (pBonf. < .001). Participants in the post-hoc-questionnaire condition reported higher WTP values (Mpost-hoc-treatment-green = 1.28, SDpost-hoc-treatment-green = 0.28; Mpost-hoc-control-green = 1.16, SDpost-hoc-control-green = 0.26) compared to those in the MRI condition (MMRI-treatment -green= 1.02, SDMRI-treatment-green = 0.35; MMRI-control-green = 1.03, SDMRI-control-green = 0.33). For the yellow color-coded frame, no significant difference between settings were found (pBonf.= .29 for control condition; pBonf.= .21 for treatment condition). There was a non-significant increase in WTP in the post-hoc-questionnaire setting (Mpost-hoc-treatment-yellow = 1.04, SDpost-hoc-treatment-yellow = 0.32, Mpost-hoc-control-yellow = 1.10, SDpost-hoc-control-yellow = 0.30) relative to the MRI setting (MMRI-treatment -yellow = 0.95, SDMRI-treatment-yellow = 0.34; MMRI-control-yellow = 1.03, SDMRI-control-yellow = 0.33). Similarly, for the red color-coded frame no significant difference between settings was found (pBonf. = .18 for treatment condition; pBonf. = .17 for control condition). Similar to the yellow-color coded condition, average WTP values were higher in the post-hoc-questionnaire setting (Mpost-hoc-treatment-red = 0.85, SDpost-hoc-treatment-red = 0.36; Mpost-hoc-control-red = 1.00, SDpost-hoc-control-red = 0.35) compared to the MRI condition (MMRI-treatment -red = 0.74, SDMRI-treatment-red = 0.33; MMRI-control-red = 0.90, SDMRI-control-red = 0.30), however not significantly different from each other.

Taken together, these data indicate no significant difference in treatment effects between the yellow and red color-coded conditions. While a significant effect was observed in the green condition (in the post-hoc questionnaire analysis), we believe this might be due to participants being allowed to consume their chosen product during the post-hoc questionnaire, thereby breaking their fast. A similar significant effect of a green traffic-light label on WTP was also reported by Enax et al. (2015), where participants had not been instructed to fast prior to the experiment. Therefore, the present findings strongly suggest that the effect of the color-coded frame observed in the rating task was similar to when participants were presented with the information of the actual Nutri-Score of the products. To also provide this information, we included an additional supplementary online S4 File, as well as a short reference in the Methods section (see Experimental Design and Behavioral Data Analysis; page 10, line 216 ff.) and the Discussion section (see Limitations, page 28, 706 ff.).

Besides the post-hoc questionnaire not included in the present manuscript, we also conducted an additional laboratory study to complete a neuroforecasting approach. The results of that add-on study are subject to another publication and therefore are not included in this paper. This data give further evidence that the color-coded frame was perceived as information and not only as a salient cue. In this laboratory study, 108 participants made 126 binary choices between the products that were shown in the presented study (63 trials in the control condition and 63 trials in the treatment condition, where the Nutri-Score label was shown). Percentage of product choice was calculated for each trial of the treatment condition of the binary choice task. These percentages served as our dependent variables. For our independent variables, behavioral and neural data from the rating task of the present study (fMRI study) were used. Specifically, for each product, we extracted the average activation from predefined voxels of interest (VOIs; using medial prefrontal cortex (mPFC) and nucleus accumbens (NAcc) of the right and left hemisphere) based on the Affect-Integration-Motivation (AIM) framework by Knutson and Genevsky (2018). We predicted product choice of the laboratory study (dependent variables) with average behavioral, neural data of the rating task (independent variables) as well as the difference in the Nutri-Score using linear regressions. We estimated three sets of linear regression models for WTP (dependent variable), and compared (1) a behavioral model, (2) a neural model, and (3) a combined model. All three models included the independent variable “Difference in Nutri-Score” to capture the effect of the label.

In the WTP domain, the behavioral model including the difference in WTP ratings and the difference in Nutri-Score explained approximately 10% of the variance in choice probability (R² = .10, p = .038). Neither predictor was significant at the 5% level, although the effect of Nutri-Score approached significance (p = .068). The neural model including the differences in activity in the right and left NAcc and mPFC, and Nutri-Score, accounted for a larger proportion of explained variance (R² = .25, p = .005). Here, the difference in Nutri-Score (p = .021) and the difference in right NAcc activation (p = .005) were significant predictors, with right NAcc activity negatively associated with the probability of choosing the left product. The combined model (behavioral, neural, and Nutri-Score predictors) slightly improved explanatory power (R² = .27, p = .006). In this model, the difference in right NAcc activation remained significant (p = .003). Neither WTP difference nor Nutri-Score difference reached significance when neural predictors were included. For detailed results see Table 1.

Table . Linear Regression Models Predicting Out-of-Sample Product Choice with FoPL from the WTP Decision Domain

Note. All standard errors are reported in parentheses. Model (1) represents the behavioral model, Model (2) the neural model, and Model (3) the combined model.

Taken together, the behavioral model did not significantly predict binary out-of-sample choices. In contrast, differences in neural activation within the selected VOIs significantly predicted out-of-sample product choice. Additionally, the difference in Nutri-Score was a significant predictor in the neural model. Since all data stem from the

---

## [Decision Letter · Decision Letter 1]

9 Sep 2025

Dear Dr. Javaheri,

Thank you for submitting your manuscript to PLOS ONE. After careful consideration, we feel that it has merit but does not fully meet PLOS ONE’s publication criteria as it currently stands. Therefore, we invite you to submit a revised version of the manuscript that addresses the points raised during the review process.

We look forward to receiving your revised manuscript.

Kind regards,

Akitoshi Ogawa, Ph.D.

Academic Editor

PLOS ONE

Journal Requirements:

Reviewers' comments:

Reviewer's Responses to Questions

**Comments to the Author**

Reviewer #1: All comments have been addressed

Reviewer #2: (No Response)

2. Is the manuscript technically sound, and do the data support the conclusions?

Reviewer #1: Yes

Reviewer #2: Yes

3. Has the statistical analysis been performed appropriately and rigorously?

Reviewer #1: Yes

Reviewer #2: Yes

4. Have the authors made all data underlying the findings in their manuscript fully available?

Reviewer #1: No

Reviewer #2: (No Response)

5. Is the manuscript presented in an intelligible fashion and written in standard English?

Reviewer #1: Yes

Reviewer #2: Yes

Reviewer #1: (No Response)

Reviewer #2: 1. I would kindly suggest formulating the hypotheses in a more formal and structured way. It would be helpful to number them and present them in a separate paragraph (e.g., H1, H2, etc.), so that you can then clearly refer back to them in the results section.

2. Regarding my earlier third comment, I feel it has not been fully addressed. As I mentioned, your study examines an affective rather than a cognitive nudge. However, references on how affective nudges influence consumer behavior are still missing. Adding these would considerably strengthen the theoretical foundation of your paper.

3. More generally, I think the theoretical background could benefit from a clearer structure. I would suggest starting with a discussion of behavioral responses to color-coded labels and frames, exploring this stream of research in more depth, and then formulating hypotheses (H1, H2, etc.) related to the behavioral aspects. Afterwards, you could discuss the literature on neural responses and also formulate formal hypotheses in this section. Following the same structure throughout the paper would make it much easier for readers to follow your argument.

4. Page 21, lines 412–414: I recommend nuancing the statement “which is consistent with prior research of how FoPL alter choices (2,19).” For the Nutri-Score in particular, findings are mixed: while some studies show that consumers tend to select more healthy products, they do not substantially modify their choices of unhealthy products (see below).

van den Akker, K., Bartelet, D., Brouwer, L., Luijpers, S., Nap, T., & Havermans, R. (2022). The impact of the nutri-score on food choice: A choice experiment in a Dutch supermarket. Appetite, 168, 105664.

De Temmerman, J., Heeremans, E., Slabbinck, H., & Vermeir, I. (2021). The impact of the Nutri-Score nutrition label on perceived healthiness and purchase intentions. Appetite, 157, 104995.

5. I recommend adding the multilevel results to the appendices, as these analyses are, in my opinion, more appropriate and provide a more accurate representation than the current approach. Including them as supplementary material would strengthen the manuscript and allow readers to better assess the robustness of the findings.

Thank you, my other comments have been addressed satisfactorily.

**Do you want your identity to be public for this peer review?** For information about this choice, including consent withdrawal, please see our Privacy Policy

Reviewer #1: No

Reviewer #2: No

---

## [Author Response · Author response to Decision Letter 2]

22 Oct 2025

Reviewer #2

1. I would kindly suggest formulating the hypotheses in a more formal and structured way. It would be helpful to number them and present them in a separate paragraph (e.g., H1, H2, etc.), so that you can then clearly refer back to them in the results section.

We thank the reviewer for this helpful suggestion. In the revised manuscript, we have reformulated our hypotheses into a more formal and structured format, numbering them as H1–H3 and presenting them in separate paragraphs in the Introduction section (p. 5, lines 82-86; p. 7, lines 132-143). This revised section now clearly distinguishes between behavioral (H1) and brain activity derived hypotheses (H2–H3), which we refer back to in the Results (e.g., p. 15, line 295) and Discussion sections (e.g., p. 21, line 425). We have additionally changed the Figures (4-6) in a way that the hypothesized regions of the neural hypotheses (H2 and H3) are better detectable.

2. Regarding my earlier third comment, I feel it has not been fully addressed. As I mentioned, your study examines an affective rather than a cognitive nudge. However, references on how affective nudges influence consumer behavior are still missing. Adding these would considerably strengthen the theoretical foundation of your paper.

We thank the reviewer for reiterating this important point. We acknowledge that the color-coded frame shows a strong affective component and cannot be considered a purely cognitive nudge. In the revised manuscript, we now explicitly mention this aspect of the Nutri-Score in the introduction chapter (p. 4, lines 56–60) and note that effect sizes on effectiveness of the nudges on making healthier food decisions can vary between different nudge classifications. Additionally, in the theoretical background of the study we further elaborate the experimental conditions of the mentioned studies, where a Guideline daily amount (GDA) label (information-based label without color code) was compared with traffic light labels (information based label with color-grading) to make it more comparable with our study, where we also used traffic-light color grading in form of a frame around food items (p. 5, lines 94 ff.). We also clarify in the limitation section (p. 29, lines 624 ff.) that our manipulation emphasized the affective (color-based) component of the Nutri-Score while preserving its cognitive content. Importantly, all participants were informed about the meaning of the frame and the Nutri-Score information before the task and post-hoc analyses (see S4 File) confirm that the frames inferred the intended evaluative information. Further, we conducted an additional neuroforecasting study that is not part of the present study but clearly demonstrated that the neural activity of the color-coded frame conditions of our fMRI sample could forecast binary choices of food products with the Nutri-Score label of an independent sample. This data shows that our manipulation did not solely rely on an emotional response, even though affective processes might have contributed to decision-making, and that the color-coded frame was associated with Nutri-Score information. Finally, we refined the research question in the introduction chapter to explicitly state that the study investigates the impact of the Nutri-Score–derived color-coded frame on three key components of dietary decision-making at the neural level. We hope that together, these changes clarify the theoretical framing and address the reviewer’s concern.

3. More generally, I think the theoretical background could benefit from a clearer structure. I would suggest starting with a discussion of behavioral responses to color-coded labels and frames, exploring this stream of research in more depth, and then formulating hypotheses (H1, H2, etc.) related to the behavioral aspects. Afterwards, you could discuss the literature on neural responses and also formulate formal hypotheses in this section. Following the same structure throughout the paper would make it much easier for readers to follow your argument.

We thank the reviewer for this valuable suggestion. In the revised manuscript, we have substantially restructured the theoretical background to improve clarity and align with the proposed organization. Specifically, we now (i) first discuss behavioral evidence on the effects of color-coded labels and frames on food choice, healthiness perception, and tastiness perception, (ii) follow this section with a clearly formulated behavioral hypothesis (H1), and (iii) subsequently present the literature on neural responses before introducing the neural hypotheses. We have also revised the presentation of the neural hypotheses: rather than grouping them by decision domain, we now group them according to the two main pathways involved, the ventral reward pathway and the cognitive control pathway. This new structure also mirrors the organization of the Results section, thereby ensuring consistency throughout the paper and making it easier for readers to follow the argument from theory to results and discussion.

4. Page 21, lines 412–414: I recommend nuancing the statement “which is consistent with prior research of how FoPL alter choices (2,19).” For the Nutri-Score in particular, findings are mixed: while some studies show that consumers tend to select more healthy products, they do not substantially modify their choices of unhealthy products (see below). van den Akker, K., Bartelet, D., Brouwer, L., Luijpers, S., Nap, T., & Havermans, R. (2022). The impact of the nutri-score on food choice: A choice experiment in a Dutch supermarket. Appetite, 168, 105664. De Temmerman, J., Heeremans, E., Slabbinck, H., & Vermeir, I. (2021). The impact of the Nutri-Score nutrition label on perceived healthiness and purchase intentions. Appetite, 157, 104995.

We thank the reviewer for this helpful suggestion. In the revised manuscript, we have nuanced our interpretation of the findings to reflect that the color-coded frame primarily discouraged unhealthy choices rather than actively promoting healthier ones, contradicting previous findings (p. 21, lines 428 ff.). We have also explicitly acknowledged that prior evidence on the Nutri-Score report effects mainly on the selection of healthier products, and we refer to the respective studies (e.g. De Temmerman et al., 2021; van den Akker et al., 2022).

5. I recommend adding the multilevel results to the appendices, as these analyses are, in my opinion, more appropriate and provide a more accurate representation than the current approach. Including them as supplementary material would strengthen the manuscript and allow readers to better assess the robustness of the findings.

We appreciate the reviewer’s suggestion and have implemented the multilevel results to the appendices (File S6). In addition, we refer to the multilevel analysis in the “Behavioral Data” section of the discussion (p. 21, line 427 ff.).

---

## [Editor Report · Decision Letter 2]

23 Oct 2025

From attributes to value: neural correlates of a front-of-package label on food decision-making – an fMRI study

PONE-D-25-19520R2

Dear Dr. Javaheri,

We’re pleased to inform you that your manuscript has been judged scientifically suitable for publication and will be formally accepted for publication once it meets all outstanding technical requirements.

Kind regards,

Akitoshi Ogawa, Ph.D.

Academic Editor

PLOS ONE
---

## [Editor Report · Acceptance letter]

PONE-D-25-19520R2

PLOS ONE

Dear Dr. Javaheri,

I'm pleased to inform you that your manuscript has been deemed suitable for publication in PLOS ONE. Congratulations! Your manuscript is now being handed over to our production team.

Kind regards,

on behalf of

Dr. Akitoshi Ogawa

Academic Editor

PLOS ONE